# Amelioration of Scopolamine-Induced Cognitive Dysfunction in Experimental Mice Using the Medicinal Plant *Salvia moorcroftiana*

**DOI:** 10.3390/brainsci12070894

**Published:** 2022-07-07

**Authors:** Fazal Wahid, Tour Jan, Fakhria A. Al-Joufi, Syed Wadood Ali Shah, Mohammad Nisar, Muhammad Zahoor

**Affiliations:** 1Department of Botany, University of Malakand, Dir (Lower), Chakdara 18800, Pakistan; fazalwahid138@yahoo.com (F.W.); mnshaalpk@yahoo.com (M.N.); 2Department of Pharmacology, College of Pharmacy, Jouf University, Sakaka 72341, Aljouf, Saudi Arabia; faaljoufi@ju.edu.sa; 3Department of Pharmacy, University of Malakand, Dir (Lower), Chakdara 18800, Pakistan; pharmacistsyed@gmail.com; 4Department of Biochemistry, University of Malakand, Dir (Lower), Chakdara 18800, Pakistan

**Keywords:** *Salvia moorcroftiana*, enzyme inhibition, antiamnesic activity, discrimination index, mice

## Abstract

*Salvia moorcroftiana* is medicinally used in various parts of the world to treat a number of diseases. In the literature, the antiamnesic activity of this plant has not yet been reported. Therefore, the current study was aimed at evaluating the in vivo antiamnesic (scopolamine-induced) potential of *Salvia moorcroftiana*. The major phytochemical groups such as total phenolic (TPC), total tannin (TTC), and total flavonoid content (TFC) in methanolic extract (SlMo-Crd) and subsequent fractions of *Salvia moorcroftiana* were quantified using standard methods. The in vitro anticholinesterase (against butyryl cholinesterase; BChE and acetylcholinesterase; AChE) and antioxidant (against 2,2-diphenyl-1-picrylhydrazyl; DPPH and 2,2′-Azino-bis (3-ethylbenzothiazoline-6-sulfonic acid); ABTS free radicals) potentials of crude (SIMO-Crd) extract and fractions (hexane; SlMo-Hex, chloroform; SlMo-Chl, ethyl acetate; SlMo-Et) were also determined. The SlMo-Crd at doses of 100 and 200 mg/kg body weight compared to fractions of 75 and 150 mg/kg body weight (which were 1/10th of the highest dose tested in acute toxicity tests) were evaluated for their memory enhancement and learning behavior in normal and scopolamine-induced mental dysfunction in mice using behavioral memory tests such as the Y-maze test and novel object recognition test (NORT). Moreover, the samples were further evaluated for acetylcholine contents and biochemical markers such as MDA (malondialdehyde), SOD (superoxide dismutase), CAT (catalase), and GSH (glutathione peroxidase) levels. The maximum TPC with a value of 114.81 ± 1.15 mg GAE/g, TTC with a value of 106.79 ± 1.07 mg GAE/g, and TFC with a value of 194.29 ± 0.83 mg RE/g were recorded for the SlMo-Chl fraction. Against the DPPH free radical, the methanolic extract exhibited an IC_50_ value of 95.29 ± 1.06 µg/mL whereas, among the fractions, the best activity was observed for the SlMo-Chl fraction with an IC_50_ of 75.02 ± 0.91 µg/mL, followed by SlMoS-Et with an IC_50_ value of 88.71 ± 0.87 µg/mL. Among the extracts, the SlMo-Chl and SlMo-Et fractions inverted the amnesic effects of scopolamine in mice effectively. Additionally, the SlMo-Chl and SIMO-Et fractions considerably enhanced the percent spontaneous alteration performance in the Y-maze test with values of 65.18 ± 2.61/69.51 ± 2.71 and 54.92 ± 2.49/60.41 ± 2.69, respectively, for the tested doses. The discrimination index (DI) in experimental mice was considerably enhanced by the SlMo-Chl in the NORT with values of 59.81 ± 1.21/61.22 ± 1.31% DI correspondingly for the tested doses, as mentioned above, followed by the SlMo-Et extract. The selected plant in the form of extracts ameliorated the effects of amnesia in mice and could, therefore, be used as a therapy for amnesia; however, this is subject to further exploration in other animal models and the isolation of the responsible compounds.

## 1. Introduction

Plants are nature’s factories, providing a wide range of diversity in terms of their chemical compositions. Plants produce two types of compounds: primary metabolites and secondary metabolites. The secondary metabolites are produced by plants to cope with environmental stress or to protect themselves from invaders, which are unique and complex compounds with a wide range of biological potential. Usually, phytochemists explore such compounds for therapeutic purposes and also it has been found that such compounds are associated with low or no side effects [1]. Medicinal plants have successfully been used by humans from time immemorial in treating a number of health complications and nowadays they are extensively explored for their therapeutic properties in treating diseases such as amnesia, dementia, and Alzheimer’s (AD). The plants have also been found to be effective in ameliorating oxidative stress, reducing brain cholinergic deficit, hyper-cholesterolaemia, and neuroinflammatory effects. They also reduce the amyloid protein deposits and thus help in restoring memory [2,3]. Neurological disorders are quite complicated and, so far, 100% efficient therapeutic drugs have not been discovered. Over the last few decades, interest in isolating antiamnesic agents from plants has increased many fold [4]. A natural compound, Huperzine A, has been reported from the Chinese medicinal plant *Huperzia serrata* (Thumb.) in 1986, which is a strong, selective, and reversible inhibitor of acetylcholinesterase (AChE) [5]. Plant extracts, isolated phytoconstituents, or their synthetic derivatives have been reported to have excellent AChE activities [6,7]. It has been claimed by many researchers that natural products can enhance memory and learning [8]. The most effective strategy to treat AD is the use of inhibitors of acetylcholinesterases [9] as these enzymes hydrolyze acetylcholine (ACh), which is needed for nerve transmission, however, a normal phenomenon in certain pathological conditions is that its excessive activities lead to cholinergic deficit on one hand whereas, on the other hand, it enhances the formation of amyloid peptide [10]. Due to this double role in AD, it became vital to explore novel effective inhibitors of AChE [11].

*Salvia moorcroftiana* is a medicinal plant belonging to the Lamiaceae family, also known as the mint family. It comprises approximately 236 genera and about 6000 species, with the largest genera being *Salvia*, *Nepeta*, *Scutellaria*, *Teucrium*, *Stachys*, *Vitex, Plectranthus*, *Hyptis*, and *Thymus*. It is a family of great diversity, with cosmopolitan distribution [12]. The largest genus of the Lamiaceae family is *Salvia* (sage), which comprises about 900 species and is native to Mediterranean regions [13]. *Salvia* is a Latin word which means “to cure”, it is used in traditional medicine by virtue of having antimicrobial [14] antitumor [15], antidiabetic [16], and antioxidant [17] potential. *Salvia* is also used as food, herbal tea [18], and as a flavoring component in cosmetics and perfumes in some parts of the world [19].

*Salvia moorcroftiana* is known by the name “Kallijari” in Pakistan. It is extensively used for the treatment of a wide range of diseases. Its seeds and roots have been used to treat cancer, piles, and coughs, whereas in the form of a bandage of leaves it is utilized in the healing of wounds and itching. Phytochemical analysis of the plant has shown the presence of flavonoid and glycoside. In a reported study, *S. moorcroftiana* has been found to have anti-inflammatory, antipyretic, and analgesic activities [20]. Other members of this genus also possess medicinal properties. *Salvia officinalis* L. has been widely used in food preparation, as a flavoring component in cosmetics, and in perfumes. Moreover, it has been used as a therapeutic agent for various diseases [21]. *Salvia fruticosa* Mill. leaves have antioxidant, anticholinesterase [22], and antimicrobial potential [23]. Similarly, *salvia hesponica* seed has been found effective in nervous system disorders, cardiovascular diseases, diabetes, and inflammatory diseases [24], while *Salvia tomentosa* Mill. leaves have been reported to have antioxidant [25,26] and antimicrobial [27] potential. The plants in this genus contain a wide range of phytoconstituents with therapeutic potential and they have the potential to be used as an antiamnesic agent.

The selected plant has not been explored medicinally as an antiamnesic agent, therefore this study is focused on investigating the memory and learning amplifying effects of *S. moorcroftiana* in scopolamine-induced amnesic mice. 

## 2. Materials and Methods

### 2.1. Chemicals

Methanol, *n*-hexane, butanol, hydrogen peroxide, chloroform, ethyl acetate, thiobarbituric acid, DPPH (2,2-diphenyl-1-picrylhydrazyl), ABTS (2,2′-Azino-bis (3-ethylbenzothiazoline-6-sulfonic acid)), donepezil, and Tween-80 were purchased from Merck (Darmstadt, Germany); BChE (lyophilized equine serum), AChE (Electric eel type-VI-S), acetylthiocholine and butyryl thiocholine iodide, DTNB and scopolamine were obtained from Sigma-Aldrich. 

### 2.2. Collection and Authentication of Plant

*S. moorcroftiana* (Wall. Ex.Benth) leaves were collected from various growing spots in the Malakand division, KPK, Pakistan. The plant was authenticated by a taxonomist at the Department of Botany, University of Malakand and a voucher specimen (BG/Sal.Mo/77-20) was placed in the herbarium of the University of Malakand.

### 2.3. Extraction and Fractionation

The samples were dried at room temperature under shade, and ground to fine powder using a mechanical grinder. The powder (4.5 kg) was soaked in 5 L of methanol and kept for 10–15 days at room temperature. Whatman filter paper and muslin cloth was used to filter the extract. The residues left was again extracted two times using fresh methanol. The filtrates were mixed and converted into a semisolid mass using a rotary evaporator at 40 °C. The crude extract (870 g) was kept in a refrigerator at 4 °C until further investigation. From the crude methanolic extract (SlMo-Crd), 800 g was further fractionated with solvents of different polarity to obtain the hexane fraction (182 g), chloroform (118 g), ethyl acetate (91 g), butanol (34 g), and aqueous (345 g) fractions.

### 2.4. Spectrophotometric Quantitative Phytochemical Analysis

The extracts of *S. moorcroftiana* were spectrophotometrically evaluated for the quantification of phytochemicals including total phenolics, flavonoids, and tannins as per the following details. 

#### 2.4.1. Estimation of the TPC

Total phenolic content in extracts was estimated using the Folin–Ciocalteu method [28]. The extracts were dissolved in 1 mg/mL in ethanol/water mixture (1:1). Then, 200 µL of each extract was mixed with 500 µL Folin–Ciocalteu reagent and finally mixed with 4.8 mL distilled water. Approximately 1 mL Na_2_CO_3_ was added to the mixture after 3 min of incubation and then 10 mL distilled water was added to dilute the sample further. The blank sample contained 200 µL of solvent instead of extract. After 1 h, the absorbance of each mixture was recorded using a spectrophotometer. A calibration curve of Gallic acid (GA) was also drawn, from which total phenolic content was estimated as mg of GA/g of dry mass of the samples. 

#### 2.4.2. Estimation of the TFC

The total flavonoid content was also estimated using a reported method in the literature [28]. The extracts were mixed with 1 mL of aluminum chloride (20 mg/mL) in ethanol, which were then further diluted to 25 mL with ethanol. Blank samples contained 1 drop of acetic acid instead of extract. After incubation for 40 min at room temperature, 200 µL of the resultant mixture was taken, and absorbance was recorded at 415 nm using a spectrophotometer. A rutin calibration curve was used to estimate the TFC and expressed as mg of RE (rutin equivalents)/g of dry mass of extract. 

#### 2.4.3. Determination of TTC

Approximately 3 mL of extracts, 1.5 mL HCl 37%, and 3.0 mL of 4% vanillin in methanol were placed in vials and incubated for 10 min in the dark. The absorbance was recorded at 500 nm using a spectrophotometer. Results were presented as mg of GA/g of dry mass of extract [29].

### 2.5. Antioxidant Activity

Antioxidant activity of the extracts was performed using ABTS and DPPH free radical scavenging assays.

The extract stock solutions was prepared by dissolving 1 mg of extracts in methanol from which then various working dilutions were prepared using dilution formula. About 2 mL of each dilution and 2 mL DPPH was mixed and incubated for 30 min at room temperature [30]. The absorbance was recorded via a spectrophotometer at 517 nm. The results were expressed in the form of IC_50_ values (as µg/mL). The percent inhibition of DPPH was estimated using Equation 1.
(1)$$\%\mathrm{DPPH}=\frac{\mathrm{Absorbance}\;\mathrm{of}\;\mathrm{control}-\mathrm{Absorbance}\;\mathrm{of}\;\mathrm{test}\;\mathrm{sample}}{\mathrm{Absorbance}\;\mathrm{of}\;\mathrm{control}}\times 100$$

Free radical scavenging activity was also evaluated using ABTS assay. Approximately 3.9 mL of ABTS solution was added to 0.1 mL of each extract dilution and incubated for 30 min. The absorbance was then recorded at 734 nm [31]. Same procedures mentioned above were followed for standards, tocopherol and ascorbic acid. 

### 2.6. In Vitro Anticholinesterase Activity

To determine the cholinesterase inhibitory potential, AChE and BChE enzymes were employed. Briefly, various dilutions of extracts (50 µL) and 0.5 mL of AChE was taken in test tubes, and incubated at 25 °C. Approximately 2.4 mL buffer and 100 µL DTNB was added to these test tubes and was again incubated for 5 min at 25 °C. The reaction was initiated by adding 40 µL pf ATChI. The reaction mixture was incubated at 25 °C for 20 min. Absorbance of each sample was recorded using a spectrophotometer at 412 nm. Similarly, the butyrylcholinesterase inhibitory potential of various dilutions of extracts was evaluated using reagents such as BChE, BTChI, DTNB and mixed as per the details given above. The absorbance of the reaction mixtures after incubation were recorded at 412 nm. Donepezil was used as standard [32].

### 2.7. Animals and Ethical Approval

Healthy Balb/C mice (8–10 weeks old and 20–23 g in weight) were procured from National Institute of Health, Islamabad, Pakistan. The mice were retained in an animal house at University of Malakand at Department of Pharmacy for acclimation. Normal pellet diet and water were provided to the animals, and they were kept for 12 h of light-dark cycle under normal laboratory conditions, having a temperature in the range of 25 ± 2 °C and humidity of 55–65%. All protocols employed were in accordance with Animal Bye-Laws 2008 (Scientific Procedures Issue-I of the Malakand University) and the UK Animals (Scientific Procedure) Act (1986), after approval from the Ethical Committee vide notification no: Pharm/EC-ChMu/34-12/20.

### 2.8. Acute Toxicity Study

Acute toxicity study of extracts was performed following the reported protocol of Lorke [33] and OECD guidelines. The extracts were evaluated in three concentrations where up to 2000 mg/kg body weight doses were found safe in cases of crude extract and 1500 mg/kg body weight for fractions. Animals were observed on a daily basis for the symptoms of diarrhea, convulsions, lethargy, sleeping, salivation, and tremor for two weeks and in case of safe doses, no toxic effects were recorded. As per defined procedure of OECD in in vivo study, one would have to test 1/10th dose of the highest safe dose, which was 200 mg/kg for crude extract and 150 mg/kg body weight for fractions. These doses were then utilized in behavioral studies [33] per the following details.

### 2.9. Antiamnesic Activity

The antiamnesic activity of crude extract and fractions of *S. moorcroftiana* was investigated using novel object recognition tests and Y-maze tests to determine the effects of extracts on learning and memory functions in mice. The apparatus was cleaned with 70% ethanol during the inter-trial interval to prevent a confounding error due to influence of odor.

#### 2.9.1. Animal Grouping and Treatment Schedule

For antiamnesic activity, the animals were randomly divided into various groups. Group I (normal control) orally received Tween-80. Group II (amnesic group) received a dose of 1mg/kg body weight scopolamine, administered intraperitoneally (i.p). Group III served as positive control and received donepezil (2 mg/kg) orally. Group IV and V received SlMo-Crd doses of 100 mg/kg and 200 mg/kg orally. Group VI to XV received oral doses mg/kg of 75 and 150 mg/kg for SlMo-Hex, SlMo-Chl, SlMo-Et, SlMo-But, and SlMo-Aq, correspondingly. The treatment was continued for 4 weeks (28 days). Scopolamine (1 mg/kg, i.p.) was administrated on the 29th day to various groups, post 30 min of the treatments, in order to study cognitive deficits in the Y-maze and novel object recognition test. All the tests were carried out between 8:00 and 12:00 a.m. to avoid variation in results due to time effects on the performance of the animals. The experimental design used is summarized in Figure 1. 

#### 2.9.2. Y-Maze Test

Y-maze test was performed following the reported methodology [34,35,36]. Y-maze is a Y shaped apparatus having three arms, arbitrarily named A, B, and C. Each arm has a length of 20 cm, with a height of 15.5 cm and 6 cm in width, oriented at an angle of 120° from the two others. Y-maze test was performed for 5 min on 29th day. Animals were located in one arm at midpoint of Y-maze apparatus, and number of arm entries, alternate arm returns (AAR) and same arm returns (SAR) were calculated. Percentage of spontaneous alternation performance (SAP) was noted using the given formula: (2)$$\%\mathrm{SAP}=\left[\frac{\mathrm{Active}\;\mathrm{alterations}\;\left(\mathrm{total}\;\mathrm{alterations}\right)}{\mathrm{possible}\;\mathrm{alterations}\;\left(\mathrm{arm}\;\mathrm{entries}\right)}-2\right]\times 100$$

#### 2.9.3. Novel Object Recognition Test

The antiamnesic activity of *S. moorcroftiana* in the form of extracts was also investigated using novel object recognition test following a reported procedure [35]. The NORT apparatus was made of plexi glass, with dimension of 40 cm × 40 cm × 30 cm. The NORT consists of sample and test phases. For habituation one day earlier than the test, the animals were allowed to explore the box freely (having no object) for 2 min. On the test day (30th day), the sample phase was conducted. In the sample phase, every mouse was positioned in open ground for 5 min with two identical objects (i.e., plastic ball). Then, the animals were reverted back to the cages. The test phase was conducted after 24 h of the sample phase in order to assess the effect of extracts on long-term memory. Every mouse was positioned once more in the open arena where the known object (plastic ball) was exchanged with a new one. The place of the object was balanced in such a way that in each group, one half of the mice on the left side of the box arena saw the new object and the other half saw the new object of the box arena on the right side to exclude sides partiality. The time (in seconds) taken for discovering every object in each phase was noted using a stopwatch. The data were recorded once the nose of mice was touching object, or at a distance of 2 cm. The time spent (in seconds) in discovering the familiar (F) object, new (N) object, and whole time (N + F) was recorded in order to estimate the percent discrimination index (DI) employing the given formula:(3)$$\%\mathrm{DI}=\frac{N-F}{N+F}\times 100$$

### 2.10. Ex Vivo Investigation for Antioxidant Enzyme Activities and Oxidative Stress Marker Level

After behavioral assessment on day 32, the animals were sacrificed by euthanasia with isoflurane in a human manner followed by extraction of the brain. A homogenate (10% w/v) in phosphate buffer (0.1 M; pH of 7.4) was prepared, which was then subjected to centrifugation. The supernatants were taken and assessed for oxidative stress biomarkers and acetylcholinesterase activities. Lipid peroxidation (LPO) was quantified by evaluating malondialdehyde (MDA) level. The level of oxidative stress biomarkers such as superoxide dismutase (SOD), catalase (CAT), and non-enzymatic parameter of glutathione (GSH) was assessed to evaluate the scopolamine effects in treated and control groups. The assessment of AChE and ACh was performed as per the reported method illustrated by Ellman et al. [37].

#### 2.10.1. Estimation of Acetylcholinesterase Level

The acetylcholinesterase level was examined in the hippocampus and frontal cortex region of the mice brain according to the reported procedure. Phosphate buffer 100 mM was used to homogenize the tissue. After incubation for 5 min, 0.1 mL of the homogenates was mixed with 2.7 mL phosphate buffer and 0.1 mL of DTNB. Then, freshly synthesized acetylthiocholine iodide (0.1 mL) was added and absorbance of the mixture was noted at 412 nm [37].

#### 2.10.2. Estimation of Superoxide Dismutase (SOD)

SOD level was determined using an ELISA kit (Sigma-Aldrich Chemie GmbH, Germany: CS0009). Brain tissue, 20 μL of enzyme solution, 200 μL of working solution and 20 μL buffer was mixed together. Then, the mixture was incubated at 37 °C for 20 min. The absorbance was noted at 450 nm using a spectrophotometer. 

#### 2.10.3. Estimation of Malondialdehyde (MDA)

Aliquots approximately 1 mL from 10% TCA and 0.5 mL distilled water was added to a 0.5 mL volume of homogenate of brain tissue, mixed well and centrifuged for 10 min at 3000 rpm. Thiobarbituric acid (TBA; 0.375%) at amount of 0.1 mL was added to 0.2 mL supernatant, kept in a water bath for 40 min at the temperature of 80ºC and then cooled to room temperature. The absorbance of the supernatant obtained after centrifugation was noted at 532 nm using a spectrophotometer [38]. 

#### 2.10.4. Estimation of Catalase Activity

For the determination of catalase activity, 1 mL of the brain homogenate of every treated group animal was added to phosphate buffer (5 mL) having pH 7.0, and then mixed with 4 mL of H_2_O_2_ solution (800 μmol). The contents of the mixture was mixed at room temperature by mild spinning motion. Then, a portion of approximately 1 mL of this reaction mixture was collected and added to 2 mL reagent dichromate/acetic. Changes in absorbance was recorded using a spectrophotometer at 570 nm, which were then converted into catalase activity as per described procedure [39].

#### 2.10.5. Estimation of GSH Concentration 

Glutathione (GSH) concentration was estimated in mice treated with scopolamine and test samples at different concentrations. The brain homogenate of approximately 400 μL was added to 80 μL of trichloroacetic acid (50%) and 320 μL of distilled water. The mixture was centrifuged at 3000 rpm for 15 min. Approximately 400 μL of the supernatant was taken and added to 20 μL of 5,5′-dithiobis (2-nitrobenzoic acid) (DTNB) 0,1M and 800 μL of Tris–HCl buffer (0.4 M, pH 8.9). The absorbance of the mixture was noted at 412 nm after 1 min of incubation. The GSH concentration was expressed in µg/g of tissue tested [39].

### 2.11. Statistical Analysis

All data were expressed as mean ± SEM (*n* = 6). The significance of variables among the f control, scopolamine-treated and extract-treated groups was estimated using Graph Pad Prism version 5.01, one-way ANOVA followed by Dunnett’s post hoc multiple tests.

## 3. Results

### 3.1. Quantitative Phytochemical Estimation for Total Phenolic, Total Tannin and Total Flavonoid Content

The TPC, TFC and TTC estimated are given in Table 1. The highest TPC value (114.81 ± 1.15 mg GAE/g) was recorded for SlMo-Chl, followed by SlMo-Et and SlMo-Crd extracts with values of 109.72 ± 0.97 and 108.20 ± 1.10 mg GAE/g, respectively. Similarly, the highest level of TTC (106.79 ± 1.07 mg GAE/g) was detected in the SlMo-Chl fraction, followed by SlMo-Et (89.93 ± 1.03 mg GAE/g) and SlMo-Crd (76.98 ± 1.01 mg GAE/g) extracts. The highest TFC was detected in the SlMo-Chl fraction (194.29 ± 0.83 mg RE/g), whereas significant amounts were also recorded in SlMo-Et (190.13 ± 0.79 mg RE/g) and SlMo-Crd (188.39 ± 0.81 mg RE/g).

### 3.2. Antioxidant Activity

Antioxidant activity of the crude extract and fractions of *S. moorcroftiana* against DPPH and ABTS radicals are given in Table 2. Against DPPH, the IC_50_ value of 95.29 ± 1.06µg/mL was recorded for the crude extract. Among the fractions, the best activity was observed for the SlMo-Chl fraction, with IC_50_ of 75.02 ± 0.91 µg/mL, followed by SlMo-Et with IC_50_ of 88.71 ± 0.87µg/mL. Standard ascorbic acid and tocopherol produced a promising inhibition of DPPH with IC_50_ of 8.21 ± 0.39µg/mL and 6.89 ± 0.31µg/mL, respectively. A similar trend in the results of the ABTS assay was observed, as observed for the DPPH assay.

### 3.3. In Vitro Cholinesterase Inhibitory Activity

The anticholinesterase potential, as determined against the AChE and BChE, is given in Appendix A. The results revealed that SlMo-Crd, SlMo-Chl and SlMo-Et extract contained potent inhibitors of these enzymes. The most potent among them was SlMo-Chl (IC_50_ of 626.92 and 547.12 μg/mL against AChE and BChE, respectively). SlMo-Et produced IC_50_ values of 759.30 and 711.77 μg/mL correspondingly against the mentioned enzymes. The standard donepezil resulted in IC_50_ values of 4.96 and 3.80 μg/mL, correspondingly, against AChE and BChE.

### 3.4. Acute Toxicity

The dose 2000 mg/kg body weight was found safe in the case of crude extract, whereas in the case of tested fraction, a 1500 mg/kg body weight dose was found safe. As per OECD guidelines, 1/10th of these doses was then given to different animal groups [40,41,42] in subsequent experiments. 

### 3.5. Evaluation of Antiamnesic Activity as Learning Behaviors

#### 3.5.1. Y-Maze Spontaneous Alteration

Y-maze spontaneous alternation test is a behavioral test used for assessing the willingness of animals to walk around new environments, as they naturally have a preference to explore a new arm of the apparatus rather than a previously visited one. Many parts of the brain are involved in this task and alteration is measured as a parameter of functional spatial memory. Figure 2 shows the findings of the Y-maze test, where scopolamine has significantly lessened the spontaneous alteration from 81.39% to 38.10% (*p* ˂ 0.001, *n* = 6). The spontaneous alteration has been normalized by donepezil, which has increased the alteration to 78.82 ± 1.96% [F (14.75) = 32.25, *p* ˂ 0.001, *n* = 6). Administration of extracts has significantly suppressed the scopolamine-induced memory deficit in the spatial memory task in a dose-dependent manner. *S. moorcroftiana* (SlMo-Crd) at doses of 100 and 200 mg/kg body weight also amplified the percent alteration. SlMo-Crd exhibited a major rise (*p* ˂ 0.01, *p* ˂ 0.001) in spontaneous alternation at 100 and 200 mg/kg in comparison to the amnesic group, which was found to be 51.68 ± 2.49% (*p* ˂ 0.01, *n* = 6) and 57.15 ± 1.95% [F (14.75) = 32.25, *p* ˂ 0.001, *n* = 6) correspondingly for the tested doses. Among the fractions, promising effects were produced by SlMo-Chl at 75 and 150 mg/kg body weight doses, compared to the amnesic group that was found to be 65.18 ± 2.61% (*p* ˂ 0.001, *n* = 6) and 69.51 ± 2.71% [F (14.75) = 32.25, *p* ˂ 0.001, *n* = 6) correspondingly. The results obtained in this study are in accordance with previously reported studies [34,35]. It could be suggested from the obtained results that the extract at the tested dose could be helpful in reinstating the memory deficit in the experimental animals.

#### 3.5.2. Novel Object Recognition Test

The NOR test is used to assess recognition memory in animal models and is based on the tendency of the experimental animal to spend more time exploring a novel object than an identical one. This exploration of the novel object reflects the learning and recognition memory level of the experimental animals. The results of the memory enhancing potential for long-term memory determined through NOR test are given in Table 3 and Appendix A. In sample phase for all tested groups, no significant variations were observed in exploring the identical objects. In the test phase, the exploration time in seconds was considerably greater for the novel object (N) than an identical one for the groups treated with crude extract (53.02 ± 1.51 and 54.16 ± 1.67% [F (14.75) = 51.79, *p* ˂ 0.001, *n* = 6) at 100, 200 mg/kg body weight), fractions and donepezil standard (69.27 ± 1.6% at 2 mg/kg body weight dose). The novel object exploring time in seconds of the donepezil group was significantly high (*p* < 0.001, *n* = 6) whereas it was lessened for the known object with a discrimination index of 69.27 ± 1.61% (*p* < 0 001, *n* = 6). Administration of the samples increased the discrimination index in the scopolamine-induced memory deficit in a dose-dependent manner. The chloroform fraction with DI of 59.81 ± 1.21 (*p* < 0 001, *n* = 6) and 61.22 ± 1.31% [F (14,75) = 51.79, *p* ˂ 0.001, *n* = 6) at 75 and 150 mg/kg body weight doses significantly increased the discrimination index compared to the amnesic group (31.43 ± 1.56%). Similarly, the SlMo-Et fraction significantly decreased the discrimination index (55.20 ± 1.72 and 57.33 ± 1.48 % [F (14,75) = 51.79, *p* ˂ 0.001, *n* = 6) at 75 and 150 mg/kg body weight doses in comparison to the amnesic group, followed by SlMo-But (discrimination index = 39.11 ± 1.61% and 40.24 ± 1.31% at 75 and 150 mg/kg body weight doses, respectively). The discrimination index of the SlMo-Aq fraction were 40.03 ± 1.53% and 41.23 ± 1.68% at 75 and 150 mg/kg body weight doses, in comparison to the amnesic group. The results of the control group, amnesic group and positive control are in accordance with previously reported studies [34,35]. The recorded observation suggests that the animals could recover from scopolamine-induced memory deficit if treated with selected plant extracts.

#### 3.5.3. Effect of Extracts on AChE Activity and ACh Levels

Administration of scopolamine caused substantial elevation of AChE activity, and decreased ACh contents in the hippocampus and frontal cortex of the experimental animal brain. The results obtained have been summarized in Appendix A.

A considerable rise in the level of AChE (Figure 3A) in the hippocampus and frontal cortex was observed after scopolamine administration (30.87 ± 1.32 and 25.61 ± 1.29, respectively, for the mentioned tested doses), which was effectively reversed by the donepezil, crude extract and fractions. The chloroform fraction at a dose of 75 mg/kg reversed the level to 16.36 ± 1.45 [F (8,45) = 5.004, *p* < 0.001, *n* = 6) and 13.37 ± 1.49 [F (8,45) = 4.255, *p* < 0.001, *n* = 6) in HC and FC, respectively. While at a dose of 150 mg/kg, a decline of 16.01 ± 1.61 and 13.01 ± 1.38 in HC and FC, respectively, was observed. Similarly, SlMo-Et caused a decrease in the level of AChE in HC and FC at a dose of 75 mg/kg body weight as 16.67 ± 1.48 and 14.49 ± 1.21, whereas at a dose of 150 mg/kg body weight the decline recorded was 16.22 ± 1.39 [F (8,45) = 5.004, *p* < 0.001, *n* = 6) and 14.11 ± 1.50 [F (8,45) = 4.255, *p* < 0.001, *n* = 6), respectively, in the mentioned organs. Simultaneously, a substantial decrease in the ACh content was also noted in the scopolamine amnesic group that was reverted back by the extract and standard (Figure 3B).

### 3.6. Assessment of Antioxidant Enzyme Activities

Administration of scopolamine caused a decrease in the CAT and SOD level in the brain of experimental animals. The crude extract, chloroform and ethyl acetate fractions showed a distinct effect on these alterations by enhancing the level of CAT [F (8,45) = 4.616, *p* < 0.001, *n* = 6) and SOD [F (8,45) = 26.53, *p* < 0.001, *n* = 6) contents, signifying the potential role as an antioxidant to ameliorate oxidative stress, as presented in Table 4.

Scopolamine administration considerably enhanced the level of MDA by 27.19 ± 1.51 units/mg of protein (*p* < 0.001, *n* = 6) in the brain homogenate, compared to the control (9.88 ± 0.98 units/mg protein, *n* = 6) group. This decline was overturned by pre-treatment with standard donepezil, which was documented as 11.29 ± 1.31 unit/mg protein [F (8,45) = 12.87, *p* < 0.001, *n* = 6) compared to the amnesic group (27.19 ± 1.51). Pre-treatment of mice with SlMo-Crd and fractions significantly enhanced the MDA level in the brain by 18.21 ± 1.39, and 18.05 ± 1.42 units/mg protein [F (8,45) = 12.87, *p* < 0.001, *n* = 6), respectively, at a dose of 100 and 200 mg/kg body weight (*p*< 0.01, *n* = 6) in comparison to the scopolamine-treated group (27.19 ± 1.51 nmol/mg). The MDA level in the brain was increased by SlMo-Chl and SlMo-Et (14.23 ± 1.38, 13.57 ± 1.67 and 16.09 ± 1.56, 15.97 ± 1.31 nmol/mg protein [F (8,45) = 12.87, *p* < 0.001, *n* = 6), respectively, at a dose of 75 and 150 kg/mg body weight).

## 4. Discussion

Plants have been used as therapeutic agents in the treatment of different ailments through the ages, being a compatible and greener source of bioactive compounds [12]. The principal phytochemical groups such as flavonoids, alkaloids, tannins, phenols, and saponins have been found to be responsible for the therapeutic potential of the plants. The members of the genus *Salvia* have been reported to contain high concentrations of TFC, TPC, and TTC. In this respect, we have estimated TFC, TPC and TTC (Table 1) where a considerable amount of these contents has been recorded and the observed biological potential might be due to these constituents. Our results are in close agreement with a reported study where high contents of TPC, TTC and TFC in the petroleum ether, chloroform, ethanol, and aqueous extracts of *Polygonatum verticillatum* have been found [43]. Oxidative stress in modern societies has increased many fold due to little or no use of vegetables and fruits, giving rise to a number of health complications such as diabetes, aging, and neurodegenerative diseases. Natural antioxidants and their health benefits have gained a lot of importance in the common population due to their low incidences of side effects [44]. Many plants have been recognized to contain natural antioxidants capable of scavenging the free radicals formed in the body, for example *Angelica pancicii*, *Achillea grandifolia, Artemisia absinthium,* and *Salvia officinalis* [44]. In the present study, the crude extract (IC_50_ = 95.29 ± 1.06 µg/mL), chloroform (IC_50_ = 75.02 ± 0.91 µg/mL), and ethyl acetate (IC_50_ = 88.71 ± 0.87 µg/mL) fractions have been found to be a potential scavenger of the DPPH free radical. The results suggest that this plant could be used as a therapeutic agent in oxidative stress and related complications. 

The cholinergic system is one of the pivotal systems that plays an important role in neurotransmission and, consequently, in learning and memory [29]. As mentioned above in AD, a decline in the level of neurotransmitter ACh is encountered due to excessive activity of AChE, leading to memory dysfunction [45]. A decline in the cholinergic depots of the body weakens the memory, whereas some plants have been found to reinstate the cholinergic level and thus improve memory [46]. A number of plants have been reported to have promising anticholinesterase potential in the form of extracts and isolated metabolites [47,48,49]. Unfortunately, AD is becoming more prevalent day by day and efforts on a huge scale are needed to discover novel drugs with prominent anti-Alzheimer potential. Dementia is the initial indicator of AD that is prevalent in aged people (25% of people above the age of 80 years). In AD patients, brain areas involved in cognitive function (e.g., the cortex and hippocampus) are affected due to the deposition of extracellular amyloid plaques [50]. Due to this disease, there is neuronal cell death in the basal forebrain [51] leading to cholinergic alteration in the hippocampus and cortex regions of the brain, causing memory loss and eventually leading to neuropsychiatric and neurological diseases [52,53,54,55,56]. In the present study, a promising inhibition of AChE and BChE has been recorded for SlMo-Crd, SlMo-Chl, and SlMo-Et (Appendix A). 

A number of medicinal plants in ancient records have been documented as memory enhancers [50]. Additionally, in the European traditional medicinal system, *Salvia* (sage) species and *Melissa officinalis* are believed to have memory-enhancing effects [57,58]. Other plants such as *Milicia excelsa* stem bark have been used as an antipsychotic agent [59,60]. In this respect, we have tested the selected plant as an antiamnesic agent in experimental animals, using a Y-maze test where mice are enthused in terms of exploration of the apparatus in a different environment. Normal mice searched out a new arm and those whose memory was not working appropriately (the amnesic group) again entered the old arm, indicating a memory dysfunction in comparison to normal mice. Despite of scopolamine treatment, for the SlMo-Crd treated groups, a significant increase in the percent spontaneous alterations were recorded (51.68 ± 2.49 and 57.15 ± 1.95% at doses of 100 and 200 mg/kg body weight, respectively), compared to the scopolamine amnesic group. The chloroform (65.18 ± 2.61 and 69.51 ± 2.71% for the tested doses) and ethyl acetate (54.92 ± 2.49 and 60.41± 2.69% for the tested doses) extracts were also equally effective. 

The antiamnesic potential was assessed in terms of both long-term and short-term memory, using the novel object recognition test. In such experiments, the normal, control and treated groups are allowed to explore the new object, which is the basic principle of this test. Normally, the animals spend more time with an unknown (novel) object compared to a known object. In the current study (Table 3), the % DI recorded for SIMo-Crd was 53.02 ± 1.51% and 54.16 ± 1.67% at doses of 100 and 200 mg/kg body weight compared to the amnesic scopolamine group (31.43 ± 1.56%) and positive control group (69.27 ± 1.61%). Furthermore, the chloroform (59.81 ± 1.21% and 61.22 ± 1.31% for the tested doses) extract was also equally effective. The study of Abhinav Kanwall [61] demonstrated that the *Vitex negundo* extract is a potent antiamnesic agent, showing that plant extracts could be used to treat such types of health complications. Similarly, Nausheen et al. [62] also documented the antiamnesic effect of fruit extracts and isolated compounds of *Elaeagnus umbellate,* using the Y-maze test and novel object recognition test. 

A strong correlation between degenerative diseases and oxidative stress has already been established. Clinically, it has been found that the generation and increase of ROS amounts encourage neuronal cell death, giving rise to oxidative stress [63]. As evidenced by a reported study, a high level of lipid peroxidation marker in the brain of the AD patient was encountered, pointing towards the involvement of oxidative stress in AD [64,65,66,67]. 

In animal models, the AChE activity and MDA level have been found to increase if treated with scopolamine and, at the same time, a decline in the level of ACh, CAT, GSH and SOD is also encountered [62,63]. Almost all extracts restored the values of the mentioned parameters in this study (Figure 3A,B), clearly indicating the antiamnesic role of this plant.

*S. moorcroftiana* extracts were found to be capable of reverting the amnesia induced by scopolamine. However, further studies in this respect are required to isolate the responsible phytoconstituents as alternative nootropic agents.

## 5. Conclusions

The crude extract and fractions of *S. moorcroftiana* possessed high contents of TFC, TPC, and TTC. The highest antioxidant activity was also recorded against ABTS and DPPH radicals for the crude extract and fractions. The extracts also ameliorate scopolamine-induced mental dysfunction in amnesic mice. In ex vivo study, the antioxidant and antiamnesic potential of the extracts were confirmed. It can be inferred from the results that *S. moorcroftiana* could be a promising natural resource that could be used as a potent neuropharmacological drug candidate against various neurodegenerative diseases. However, further exploration in this respect is needed to isolate pure pharmacologically active compounds responsible for the neuroprotective effect observed.

## Figures and Tables

**Figure 1 brainsci-12-00894-f001:**
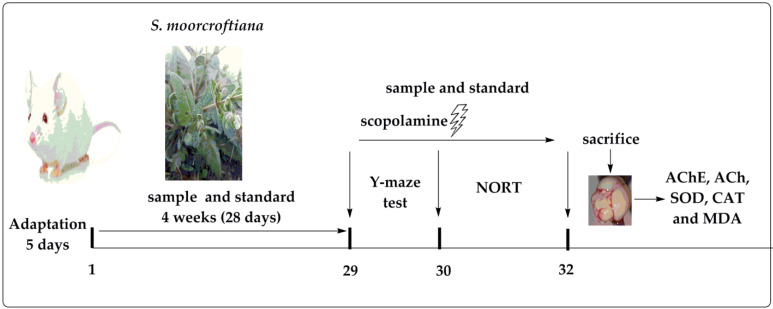
Experimental design for the assessment of memory in scopolamine-induced amnesic mice.

**Figure 2 brainsci-12-00894-f002:**
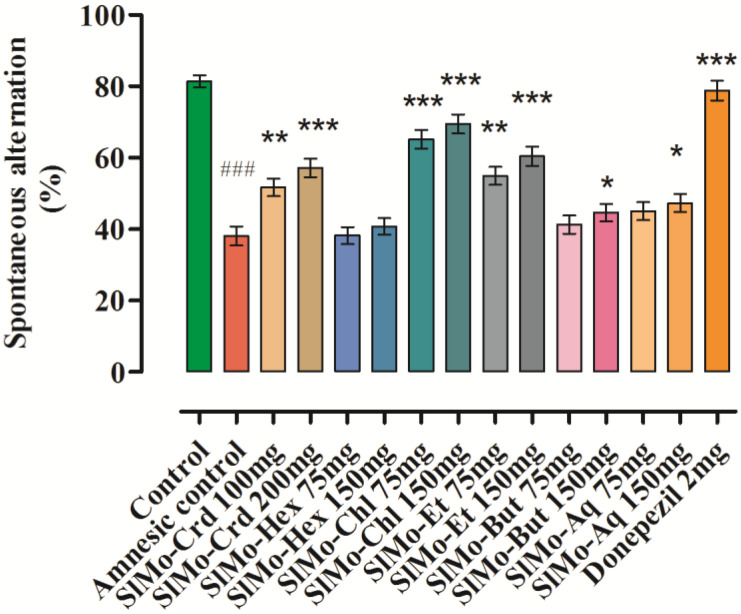
Effect of crude methanolic extract and fractions of *S. moorcroftiana* on mice for memory in behavioral Y-maze test. Mean ± SEM (*n* = 6). One-way ANOVA followed Dunnett’s post hoc multiple comparison test to calculate the values of *p*. ### *p* ˂ 0.001 is comparison of scopolamine-treated (amnesic) group with normal control, * *p* ˂ 0.05, ** *p* ˂ 0.01 and *** *p* ˂ 0.001 represents comparison of scopolamine-treated (amnesic) group vs Donepezil, crude extract and fractions-treated groups.

**Figure 3 brainsci-12-00894-f003:**
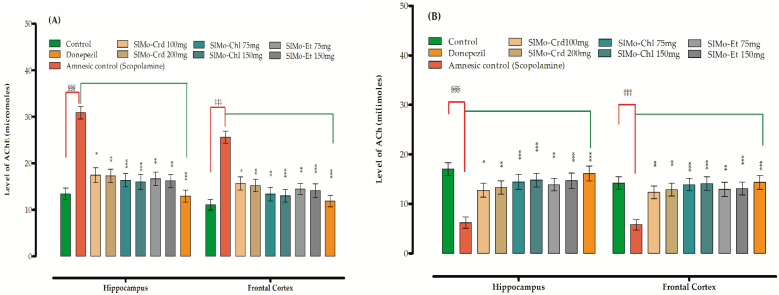
Effect of SlMo-Crd and fractions on (**A**) AChE and (**B**) ACh level in brain. Mean ± SEM (*n* = 6). Oneway ANOVA followed by Dunnett’s post hoc multiple comparison test have been used to estimate the values of *p*; ^§§§^ *p* < 0 001, ^‡‡‡^
*p* < 0 001 has been used for comparison of scopolamine- treated (amnesic) group vs normal control group, * *p* < 0.05, ***p* < 0.01 and ****p* < 0.001 compare the scopolamine-treated (amnesic) group vs Donepezil, crude extract and fractions-treated groups.

**Table 1 brainsci-12-00894-t001:** Quantitative phytochemical analysis of *S. moorcroftiana*.

Samples	TPC (mg GAE/g)	TFC (mg RE/g)	TTC (mg GAE/g)
SlMo-Crd	108.20 ± 1.10	188.39 ± 0.81	76.98 ± 1.01
SlMo-Hex	33.71 ± 1.01	36.26 ± 0.56	28.11 ± 0.70
SlMo-Chl	114.81 ± 1.15	194.29 ± 0.83	106.79 ± 1.07
SlMo-Et	109.72 ± 0.97	190.13 ± 0.79	89.93 ± 1.03
SlMo-Bt	44.31 ± 0.81	39.62 ± 0.77	29.73 ± 0.67
SlMo-Aq	46.48 ± 0.73	40.74 ± 0.62	22.89 ± 0.56

All values are determined as mean ± SEM, *n* = 3, TPC: total phenolic content, TTC: total flavonoid content, TFC: total tannin content. SlMo-Crd: crude extract, SIMo-Hex: hexane fraction, SlMo-Chl: chloroform fraction, SlMo-Et: ethyl acetate fraction, SlMo-Bt: butanol fraction, SlMo-Aq: aqueous fraction.

**Table 2 brainsci-12-00894-t002:** Antioxidant potential extracts of *S. moorcroftiana*.

Test Sample		Antioxidant Activity IC_50_(µg/mL)	
		(DPPH)	(ABTS)
*S. moorcroftiana*	SlMo-Crd	95.29 ± 1.06	120.11 ± 1.01
	SlMo-Hex	221.84 ± 1.05	249.73 ± 1.13
	SlMo-Chl	75.02 ± 0.91	90.57 ± 0.91
	SlMo-Et	88.71 ± 0.87	94.21 ± 0.81
	SlMo-Bt	137.62 ± 0.95	176.07 ± 1.01
	SlMo-Aq	164.95 ± 0.91	167.23 ± 1.04
Standard	Ascorbic Acid	8.21 ± 0.39	10.04 ± 0.41
	Tochopherol	6.89 ± 0.31	5.36 ± 0.29

All values are presented as mean ± SEM, *n* = 3, SlMo-Crd: crude extract, SlMo-Chl: chloroform fraction, SlMo-Et: ethyl acetate SlMo-Bt: butanol and SlMo-Aq: aqueous fraction of *S. moorcroftiana*.

**Table 3 brainsci-12-00894-t003:** Effects of crude extract and various fractions of *S. moorcroftiana* on mice in behavioral NORT on long-term memory.

Treatment/Dose (mg)		Sample Phase		Test Phase		DI (%)
		Identical Object A1	Identical Object A2	Novel Object A1	Familiar Object A2	
Control		19.88 ± 1.01	20.92 ± 0.91	13.17 ± 0.92	11.89 ± 0.85	52.55 ± 1.67
Amnesic control (Scopolamine)		12.91 ± 0.77	11.70 ± 0.90	6.13 ± 0.47	13.37 ± 0.73	31.43 ± 1.56
SlMo-Crd	100	18.31 ± 1.02	17.14 ± 0.97	12.46 ± 0.93 **	11.04 ± 0.81	53.02 ± 1.51
	200	17.98 ± 0.81	18.96 ± 0.83	14.05 ± 0.87 ***	11.89 ± 0.83	54.16 ± 1.67
SlMo-Hex	75	18.75 ± 0.79	17.61 ± 0.92	10.88 ± 0.84 **	17.30 ± 0.93	38.61 ± 1.57
	150	17.34 ± 0.92	19.11 ± 1.01	10.95 ± 0.89 ***	16.54 ± 0.91	39.83 ± 1.49
SlMo-Chl	75	18.47 ± 0.70	17.56 ± 0.89	16.43 ± 0.77 **	11.02 ± 0.72	59.81 ± 1.21
	150	18.09 ± 0.79	18.97 ± 0.85	17.45 ± 0.70 ***	11.05 ± 0.95	61.22 ± 1.31
SlMo-Et	75	19.02 ± 1.02	18.15 ± 0.98	13.57 ± 0.81 **	11.01 ± 0.78	55.20 ± 1.72
	150	17.87 ± 0.93	19.02 ± 0.83	14.76 ± 0.67 ***	10.98 ± 0.82	57.33 ± 1.48
SlMo-But	75	18.24 ± 1.04	17.87 ± 0.98	9.81 ± 0.65 **	15.28 ± 0.81	39.11 ± 1.61
	150	17.89 ± 0.97	18.92 ± 0.91	10.15 ± 0.80 ***	15.07 ± 0.62	40.24 ± 1.31
SlMo-Aq	75	18.75 ± 1.05	17.69 ± 0.90	10.16 ± 0.73 **	15.22 ± 0.96	40.03 ± 1.53
	150	17.89 ± 0.96	19.01 ± 1.06	10.40 ± 0.78 ***	14.82 ± 0.93	41.23 ± 1.68
Donepezil	2	17.55 ± 0.99	18.07 ± 0.93	23.47 ± 0.90 ***	10.41 ± 0.85	69.27 ± 1.61

Mean ± SEM (*n* = 6). One-way ANOVA followed by Dunnett’s post hoc multiple comparison test has been used to estimate the values of *p*; ** *p* ˂ 0.01 and *** *p* ˂ 0.001.

**Table 4 brainsci-12-00894-t004:** Effects of extracts on different biomarker levels in brain of mice.

Sample Test (mg)		SOD (U/mg of Protein)	CAT (U/mg of Protein)	MDA (nmol/mg Protein)	GSH (μg/mg of Protein)
Control		14.29 ± 1.29	31.22 ± 1.49	9.88 ± 0.98	46.91 ± 1.54
Amnesic control (Scopolamine)		5.21 ± 0.87 ###	7.12 ± 1.22 ###	27.19 ± 1.51 ###	14.82 ± 1.71 ###
SlMo-Crd	100	7.21 ± 1.33 **	22.09 ± 1.41 **	18.21 ± 1.39 ***	33.02 ± 1.33 **
	200	8.16 ± 1.21 **	23.11 ± 1.23 **	18.05 ± 1.42 **	34.11 ± 1.49 **
SlMo-Chl	75	8.89 ± 1.41 ***	24.03 ± 1.39 ***	14.23 ± 1.38 **	36.47 ± 1.53 ***
	150	9.46 ± 1.67 ***	25.37 ± 1.42 ***	13.57 ± 1.67 ***	39.13 ± 1.38 ***
SlMo-Et	75	8.06 ± 1.12 **	24.27 ± 1.57 **	16.09 ± 1.56 **	34.69 ± 1.30 **
	150	8.91 ± 1.21 ***	24.51 ± 1.39 ***	15.97 ± 1.31 **	35.89 ± 1.46 ***
Donepezil	2	12.87 ± 1.39 ***	32.01 ± 1.31 ***	11.29 ± 1.31 ***	46.81 ± 1.43 **

Mean ± SEM (*n* = 6). One-way ANOVA followed Dunnett’s post hoc multiple comparison test have been used to estimate the values of *p*. ### *p* < 0 001 represents a comparison of scopolamine-treated (amnesic) group vs normal control, ** *p* < 0.01 and *** *p* < 0.001 are comparison of scopolamine-treated (amnesic) group with Donepezil, crude extract and fractions-treated groups.

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
