# Peer review of "Amelioration of Scopolamine-Induced Cognitive Dysfunction in Experimental Mice Using the Medicinal Plant Salvia moorcroftiana"

_brainsci, 2022, doi:10.3390/brainsci12070894_

Round 1

Reviewer 1 Report

Authors bring interesting findings on nootropic effects of Salvia moorcroftiana extract.

Experiments are properly designed and conducted, and results are plausible, having been clearly presented.

There are only a few minor comments:

  1. What strain of "albino mice" was used? (was it balb/c strain?), Please declare.
  2. Please explain why tween-80 served as a control substance for in vivo experiments? Was it solvent for active substances also? Please state why tween-80 was chosen.

Author Response

Reviewer 1:

Authors bring interesting findings on nootropic effects of Salvia moorcroftiana extract.

Experiments are properly designed and conducted, and results are plausible, having been clearly presented.

  • Thank you worthy reviewer for encouraging remarks

There are only a few minor comments:

Reviewer question #1:  What strain of "albino mice" was used? (was it balb/c strain?), Please declare.

  • Author response:Worthy reviewer, thanks for valuable suggestion. The statement has been corrected accordingly and quoted as Healthy Balb/C mice aged 8–10 weeks old (20–23 g)  in the revised paper

Reviewer question #1: Please explain why tween-80 served as a control substance for in vivo experiments? Was it solvent for active substances also? Please state why tween-80 was chosen.

  • Author response:Worthy reviewer, thanks for valuable suggestion. Tween-80 is used for administration as a medium for suspension preparation when the sample is not freely soluble in normal saline. That is why the samples were suspended in 2% Tween-80 in normal saline. Also, in literature this is frequently used. Few of the references are given below
  • Ortega, Rocio, Miguel Valdés, Francisco J. Alarcón-Aguilar, Ángeles Fortis-Barrera, Elizabeth Barbosa, Claudia Velazquez, and Fernando Calzada. "Antihyperglycemic Effects of Salvia polystachya Cav. and Its Terpenoids: α-Glucosidase and SGLT1 Inhibitors." Plants 11, no. 5 (2022): 575.
  • Merlín-Lucas, Verenice, Rosa María Ordoñez-Razo, Fernando Calzada, Aida Solís, Normand García-Hernández, Elizabeth Barbosa, and Miguel Valdés. "Antitumor potential of annona muricata linn. An edible and medicinal plant in mexico: In vitro, in vivo, and toxicological studies." Molecules 26, no. 24 (2021): 7675.
  • Sirak, Betelhem, Lea Mann, Adrian Richter, Kaleab Asres, and Peter Imming. "In vivo antimalarial activity of leaf extracts and a major compound isolated from Ranunculus multifidus Forsk." Molecules 26, no. 20 (2021): 6179.

Reviewer 2 Report

This study investigated the amelioration of scopolamine induced cognitive dysfunction in experimental mice using the medicinal plant, Salvia moorcroftiana. Although the authors demonstrated some interesting findings, the following points should be improved to strengthen the authors' assertion.

1. The innovation of the manuscript is not enough , and there are no outstanding innovations in the manuscript.

2. The figures in the manuscript are not standardized and the figuress are presented in different ways. For example, figure S1 "()%" in the supplementary materials.

3. We suggested that the author should supplement experiments in the manuscript to determine the chemical structure formula that plays a major role.

4. The behavioral memory model test results in the manuscript experiment results need to be further improved.

5. The discussion part should be rewritten and needs to be significantly improved to clarify what is new in light of existing literature.

Author Response

Reviewer 2:

This study investigated the amelioration of scopolamine induced cognitive dysfunction in experimental mice using the medicinal plant, Salvia moorcroftiana. Although the authors demonstrated some interesting findings, the following points should be improved to strengthen the authors' assertion.

  • Thank you worthy reviewer, for the positive input.

Reviewer question #1: The innovation of the manuscript is not enough, and there are no outstanding innovations in the manuscript.

  • Author response:Worthy reviewer, novelty or innovation are only the terms but have no physical existence and using such terms a paper can easily be rejected without having a solid proof. It depends and vary from person to person. For example, you report very genuine research and as reviewer I says it is not novel then who will decide, as novelty do not have a physical existence. Here in this case worthy reviewer, you commented about innovation it would be better you elaborate your point like, is this plant has used before for such activities? If yes then please provide that reference. To best of our knowledge this plant has not been tested/reported for such types of biological potentials. Worthy reviewer, so far, I have published 197 papers in reputed journal and constantly I am reviewing for journal but I have never commented on novelty instead I am commenting on technicality.

Reviewer question #2: The figures in the manuscript are not standardized and the figuress are presented in different ways. For example, figure S1 "()%" in the supplementary materials.

  • Author response:Worthy reviewer, thanks for valuable suggestion. The figure S1 in the supplementary materials has been corrected as "(%)"

Reviewer question #3:  We suggested that the author should supplement experiments in the manuscript to determine the chemical structure formula that plays a major role.

  • Author response:Worthy reviewer, thanks for valuable suggestion. However, our work is about extracts not about isolated compounds. Formulae are given in that case if you have isolated and purified them. Extracts are mixture of compounds and writing chemical formulae for such extract is not appropriate. The research on the plant is underway; the secondary metabolites have been isolated in pure form that is under investigation for characterization and structure elucidation. These characterized compounds will be subjected for the same activities.

Reviewer question #4: The behavioral memory model test results in the manuscript experiment results need to be further improved.

Author response: Worthy reviewer, thanks for valuable suggestion. The behavioral memory model test results have been improved accordingly as presented in section 3.5.1. Y-maze spontaneous Alteration and section 3.5.2. Novel Object recognition test

Reviewer question #5: The discussion part should be rewritten and needs to be significantly improved to clarify what is new in light of existing literature.

Author response: Worthy reviewer, what is new has already mentioned in point one. Discussion was improved accordingly. 

This manuscript is a resubmission of an earlier submission. The following is a list of the peer review reports and author responses from that submission.

Round 1

Reviewer 1 Report

The manuscript entitled “Amelioration of scopolamine induced cognitive dysfunction in experimental mice using the medicinal plant, Salvia moorcroftiana” by Zahoor is focusing the potential role of Salvia moorcroftiana in the treatment of cognitive dysfunction induced by scopolamine. The topic of the manuscript is very interesting and highly relevant. The evaluation of medicinal plants, as a potential therapeutic approach, in the cure of various neurophysiological disorders is very actual for at least a couple of years, but still, there is a need for the new natural compounds. The subject area is covered by appropriate references but still needs to be improved by employing some more specific literature data. The manuscript should be better organized according to the Journal’s standards, which will make it clearer and easier to follow. The conclusions are almost consistent with the evidence, but some additional arguments from similar studies should be included to address the main question.

There are some serious concerns:

List of authors is incomplete.

Abstract is too long.

In MM section indicate precisely what are the parameters for lipid peroxidation, for antioxidant enzymes, and non-enzymatic mechanisms.

Table 1: reorder A, B, C in Legend

Statistical analysis section is missing.

What software has been used in this study?

Can you provide more parameters from NOR test?

Author Response

Report 1

Comments and Suggestions for Authors

The manuscript entitled “Amelioration of scopolamine induced cognitive dysfunction in experimental mice using the medicinal plant, Salvia moorcroftiana” by Zahoor is focusing the potential role of Salvia moorcroftiana in the treatment of cognitive dysfunction induced by scopolamine. The topic of the manuscript is very interesting and highly relevant. The evaluation of medicinal plants, as a potential therapeutic approach, in the cure of various neurophysiological disorders is very actual for at least a couple of years, but still, there is a need for the new natural compounds. The subject area is covered by appropriate references but still needs to be improved by employing some more specific literature data. The manuscript should be better organized according to the Journal’s standards, which will make it clearer and easier to follow. The conclusions are almost consistent with the evidence, but some additional arguments from similar studies should be included to address the main question.

There are some serious concerns:

Reviewer 1:

Reviewer question #1:  List of authors is incomplete.

Author response: Worthy reviewer, the word ‘and’ at the end on of authors list was added by mistake. Thanks for pointing out the mistake. The word ‘and’ was moved to its suitable position.

Reviewer question #2:  Abstract is too long.

Author response: Worthy reviewer, the abstract has been remodified and made short. However, in MDPI journals there is no such restrictions on the length of abstracts.

Reviewer question #3:  In MM section indicate precisely what are the parameters for lipid peroxidation, for antioxidant enzymes, and non-enzymatic mechanisms.

Author response: Worthy reviewer, the required parameters were accordingly mentioned in section 2.10. hopefully it will be ok now.

Reviewer question #4:  Table 1: reorder A, B, C in Legend

Author response: Worthy reviewer, in Table 1 the legends were mistakenly numbered as A, B, C which was incorrect and were deleted. The parameters were arranged according to sequence of the study.

Reviewer question #5:  Statistical analysis section is missing.

Author response: Worthy reviewer, the statistical analysis section has been added as per following details:

2.11. Statistical Analysis

All data were expressed as mean±SEM (n=6). The significance of difference among the values of control, scopolamine treated standard drug and extract treated groups for each session using Graph Pad Prism version 5.01 was determined by ANOVA (one-way) followed by Dunnett’s post hoc multiple test.

Reviewer question #6:  What software has been used in this study?

Author response: Worthy reviewer, the name of software has been added as per valuable suggestion. Graph Pad Prism version 5.01

Reviewer question #6:  Can you provide more parameters from NOR test?

Author response: Worthy reviewer, thanks for investigating the parameters for NORT. Most parameters are already mention in the material and method section whereas some are

The Y-maze apparatus used is made of plywood having white color with (40 cm × 40 cm × 66 cm) having a network floor that could be smoothly washed (by Ethanol at 70%v/v) after each test.

Reviewer 2 Report

The manuscript entitled Amelioration of scopolamine induced cognitive dysfunction in experimental mice using the medicinal plant, Salvia moorcroftiana by Fazal Wahid and collaborators evaluates the beneficial effects of Salvia moorcroftiana methanolic extract and its fractions in an mice amnesia model induced by scopolamine administration. There are some parts of all the manuscript that in my opinion need clarification. The comments are described below:

  1. The introduction section is mostly based on articles older than 10 years, only two or three articles cited by the authors being published in the last 5 years. The first introduction paragraph seems to summarize the importance of medicinal plants as therapeutic agents for different disorder related to nervous system. The second introduction paragraph present general information regarding Lamiaceae family and Salvia genus and certain therapeutic properties of some Salvia species. However, there are no mentions about the already known therapeutic properties or other information or description regarding S. moorcroftiana. The S. moorcroftiana is mentioned only once throughout the whole introduction section and the information provided by the authors are not supported by any citations.
  2. In the Materials and Methods section, please mention whether the whole plant has been used for the extraction procedure or just specific parts (e.g., flowers or roots).
  3. In the Materials and Methods section, for Estimation of TPC, please mention the number of replicates.
  4. The toxicity of S. moorcroftiana methanolic extract and its fractions has been tested directly in vivo, without any in vitro preliminary evaluation and this fact might raise ethical concerns.
  5. In the Materials and Methods section, line 229, please adjust the subsection title. Y-maze, as the authors also stated, is used for memory evaluation, but the spontaneous alternation is actually a parameter within this test. Therefore, the subsection title, in this form, is incorrect.
  6. In the Materials and Methods section, 2.8.1. subsection, the authors stated, at line 235, that the animal was placed in one of the maze arms, but at line 242 they mention that the animal was placed at the midpoint of the maze.
  7. In the Materials and Methods section, please revise the 2.8.2. subsection. There are a lot of inconsistencies in this subsection.
  8. In the Materials and Methods section, please insert a subsection to explain the method used for animal humane killing and the method used to obtain brain homogenates for further analysis.
  9. In the Materials and Methods section, 2.9. subsection, please rephrase” Lipid peroxidation (LPO) was quantified...” (line 278). Only MDA is a marker of lipid peroxidation.
  10. In the Materials and Methods section, 2.9.2. subsection, please mention the manufacturer for the SOD assay kit.
  11. All the formulas presented in the Materials and Methods section appear as blurred. Please revise this aspect.
  12. For the Results section, please consider moving the tables in the supplementary files since the relevant data are already presented in the text.
  13. In the Results section, 3.3. subsection, please mention if animals presented any side effects as a result of S. moorcroftiana methanolic extract and its fractions administrations.
  14. In the Results section, Figure 2, the spontaneous alternation should be expressed as percent, as the formula presented in Materials and Methods section indicates.
  15. In the Results section, for the subsections 3.4.2., 3.5. and 3.6., please consider to present the obtained data as graphs and to move the tables in supplementary files.
  16. In the Results section, for all utilized positive controls and also for the amnesic control, please align the findings from this manuscript to the existing literature.
  17. Most of the Discussion section actually recapitulate the information already presented in the Results section.
  18. Throughout Materials and Methods, Results and Discussion section, the authors refer to the behavioral tests as being used to asses antiamnesic activity (line 186), learning behaviors (line367) and memory performance (lines 545 and 555). Please check this aspect and use the appropriate description regarding the behavioral tests purpose.
  19. Please check the numeration of the subsections in the Materials and Methods and Results sections.
  20. Please explain all the abbreviation utilized in the manuscript.

Author Response

Report 2

Comments and Suggestions for Authors

The manuscript entitled Amelioration of scopolamine induced cognitive dysfunction in experimental mice using the medicinal plant, Salvia moorcroftiana by Fazal Wahid and collaborators evaluates the beneficial effects of Salvia moorcroftiana methanolic extract and its fractions in an mice amnesia model induced by scopolamine administration. There are some parts of all the manuscript that in my opinion need clarification. The comments are described below:

  • Worthy reviewer, thank you for your positive input, We have tried our level best to incorporate all your valuable suggestions into the revised paper.

Reviewer question #1: The introduction section is mostly based on articles older than 10 years, only two or three articles cited by the authors being published in the last 5 years. The first introduction paragraph seems to summarize the importance of medicinal plants as therapeutic agents for different disorder related to nervous system. The second introduction paragraph present general information regarding Lamiaceae family and Salvia genus and certain therapeutic properties of some Salvia species. However, there are no mentions about the already known therapeutic properties or other information or description regarding S. moorcroftiana. The S. moorcroftiana is mentioned only once throughout the whole introduction section and the information provided by the authors are not supported by any citations.

Author response: Worthy reviewer, thanks for valuable suggestion. The older references have been replaced new references published in last 5 years. The whole section has been rephrased with proper connectivity among the information presented. Hope it will ok now.

Reviewer question #2: In the Materials and Methods section, please mention whether the whole plant has been used for the extraction procedure or just specific parts (e.g., flowers or roots).

Author response: Worthy reviewer, the leaves of selected plant has been added which has clearly mentioned in the respective section.  

Reviewer question #3: In the Materials and Methods section, for Estimation of TPC, please mention the number of replicates.

Author response: Worthy reviewer, It is quoted as All the experiments were performed in triplicates, in the revised manuscript

Reviewer question #4: The toxicity of S. moorcroftiana methanolic extract and its fractions has been tested directly in vivo, without any in vitro preliminary evaluation and this fact might raise ethical concerns.

Author response: Worthy reviewer, thanks for valuable suggestion. The information has been added as per following details. “3.4. acute toxicity study” and Selection of doses for in-vivo pharmacological assessment of cognitive function using animal model.

According to OECD guide lines first we evaluated the extracts in three concentrations where up to 2000 mg/kg body weight doses were found safe in case of crude extract and 1500 mg/kg body weight for fraction. Animals were checked on daily basis for observing signs of diarrhea, convulsions, lethargy, sleeping, salivation and tremor for two weeks and no toxic effects were found. As per defined procedure of OECD in in vivo study one would test the highest dose as 1/10th of the highest safe dose which was 200 mg/kg for extract and 150 mg/kg for fractions were selected for behavioral studies after preliminary pharmacological assessment in our laboratory as well as published data elsewhere

Reviewer question #5: In the Materials and Methods section, line 229, please adjust the subsection title. Y-maze, as the authors also stated, is used for memory evaluation, but the spontaneous alternation is actually a parameter within this test. Therefore, the subsection title, in this form, is incorrect.

Author response: Worthy reviewer, the subsection has modified as Y- Maze test for spontaneous alternation. Thank you for your valuable suggestion.

Reviewer question #6: In the Materials and Methods section, 2.8.1. subsection, the authors stated, at line 235, that the animal was placed in one of the maze arms, but at line 242 they mention that the animal was placed at the midpoint of the maze.

Author response: Worthy reviewer, the statement has been remodified as Animals were located in one arm at midpoint of Y-maze apparatus.

Reviewer question #7: In the Materials and Methods section, please revise the 2.8.2. subsection. There are a lot of inconsistencies in this subsection.

Author response: Worthy reviewer, this section has been remodified as per valuable suggestion as

Y-Maze test was carried according reported methodology in order  to estimate the memory of the  experimental mice [33]. Y-maze is a Y shaped apparatus having three arms and size of each arm is equivalent in size, and were considered as A, B, and C. Each arm has length of 20 cm, with height of 15.5 cm and 6 cm in width, was oriented at an angle of 120° from the two others. Y-maze test was performed for 5 min on each animal. Animals were located in one arm at midpoint of Y-maze apparatus, and sum of arm entries and order were noted. Arm entry was determined to be a complete one after mice’s hind paw was totally inside of a specific arm; however an alternation was stated as a repeated entry of a mouse into the various arm of the y-maze apparatus. To study percent spontaneous alteration behavior, every mouse was employed at midpoint of Y-Maze apparatus and endorsed to walk easily over the maze on 29th day of study. Sequence of the arm entries was visually noted, To initiate test for behavioral study the animal was kept in one arm and the series of arm entries were documented, alternate arm returns (AAR) and same arm returns (SAR), were calculated. Percentage of spontaneous alternation performance (SAP) was noted by the given formula:

Reviewer question #8: In the Materials and Methods section, please insert a subsection to explain the method used for animal humane killing and the method used to obtain brain homogenates for further analysis.

Author response: Worthy reviewer, this section has been remodified and method for animal humane killing has been added as per valuable suggestion. The statement is presented as

After behavioral assessment on day 32nd, the animals were sacrificed by euthanasia with isoflurane in a humane manner followed by extraction of the brain.

Reviewer question #9: In the Materials and Methods section, 2.9. subsection, please rephrase” Lipid peroxidation (LPO) was quantified...” (line 278). Only MDA is a marker of lipid peroxidation.

Author response: Worthy reviewer, thanks for valuable suggestion. The statement has been modified as

Lipid peroxidation (LPO) was quantified by evaluating Malondialdehyde (MDA). The level of superoxide dismutase (SOD), catalase (CAT) and glutathione (GSH), as oxidative stress markers was also assessed.

Reviewer question #10: In the Materials and Methods section, 2.9.2. subsection, please mention the manufacturer for the SOD assay kit.

Author response: Worthy reviewer, the name of Kit assay has been added accordingly.

Reviewer question #11: All the formulas presented in the Materials and Methods section appear as blurred. Please revise this aspect.

Author response: Worthy reviewer, all the formulas have been corrected as suggested.

Reviewer question #12: For the Results section, please consider moving the tables in the supplementary files since the relevant data are already presented in the text.

Author response: Worthy reviewer, Table 3 and Table 5 have been made as supplementary tables. Now these tables are presented as Table S1 and Table S2.

Reviewer question #13: In the Results section, 3.3. subsection, please mention if animals presented any side effects as a result of S. moorcroftiana methanolic extract and its fractions administrations.

Author response: Worthy reviewer, the statement has been remodified in subsection 2.8 as

Animals were checked on daily basis for observing signs of diarrhea, convulsions, lethargy, sleeping, salivation and tremor for two weeks. The animals were also monitored for mortality if occur.

The statement in subsection 3.4 has been added as

Animals were checked on daily basis for observing signs of diarrhea, convulsions, lethargy, sleeping, salivation and tremor for two weeks and no toxic effects were found.

Reviewer question #14: In the Results section, Figure 2, the spontaneous alternation should be expressed as percent, as the formula presented in Materials and Methods section indicates.

Author response: Worthy reviewer, the Figure 2 has been modified as per valuable suggestion.

Reviewer question #15: In the Results section, for the subsections 3.4.2., 3.5. and 3.6., please consider to present the obtained data as graphs and to move the tables in supplementary files.

Author response: Worthy reviewer, Table 3 and Table 5 has been moved to supplementary file as Table S1 and Table S2 as per valuable suggestion.

Reviewer question #16: In the Results section, for all utilized positive controls and also for the amnesic control, please align the findings from this manuscript to the existing literature.

Author response: Worthy reviewer, in journals where results are presented as separate sections. Citation of references are restricted to a minimum number. Also the results are not compared with reported studies in literature. They are only compared in discussion section. In our discussion section such information are already there. Still for your satisfaction the following statement has been made in the respective section.

The results of control group, amnesic group and positive control standard obtained from animals in this study are in accordance to previous reported literature [34,35].

Reviewer question #17: Most of the Discussion section actually recapitulate the information already presented in the Results section.

Author response: Worthy reviewer,

Reviewer question #18: Throughout Materials and Methods, Results and Discussion section, the authors refer to the behavioral tests as being used to asses antiamnesic activity (line 186), learning behaviors (line367) and memory performance (lines 545 and 555). Please check this aspect and use the appropriate description regarding the behavioral tests purpose.

Author response: Worthy reviewer, the statements in respective lines as suggested has been remedified as per valuable suggestion and quoted a

  • Evaluation of antiamnesic activity for learning behaviors
  • To conduct study on antimnesic potential of plant using animal, Y- maze spontaneous alteration test is used.
  • The antiamnesic potentials assessed in terms of both long term and short term memory, the novel object recognition test was used.

Reviewer question #19: Please check the numeration of the subsections in the Materials and Methods and Results sections.

Author response: Worthy reviewer, the numbers has been corrected accordingly

Reviewer question #20: Please explain all the abbreviation utilized in the manuscript.

Author response: Worthy reviewer, all abbreviations have been explained in the manuscript as per valuable suggestion.

Round 2

Reviewer 1 Report

The authors failed to respond adequately to the required tasks.

1.

My remark: What software has been used in this study?

Author response: Worthy reviewer, the name of the software has been added as per valuable suggestion. Graph Pad Prism version 5.01.

My comment: this is not for sure software for behavioral tests analyses. Even more, it is a well- known program for statistical analysis and presentation.

2.

My remark: Can you provide more parameters from NOR test?

Author response: Worthy reviewer, thanks for investigating the parameters for NORT. Most parameters are already mentioned in the material and method section whereas some are

The Y-maze apparatus used is made of plywood having white color with (40 cm × 40 cm × 66 cm) having a network floor that could be smoothly washed (by Ethanol at 70%v/v) after each test.

My comment: Even incomplete (pay attention to the sentence), the response is not pointing my remark considering NOR test parameters. Instead, the authors offered the description of a completely different test.

Reviewer 2 Report

The authors of the manuscript entitled Amelioration of scopolamine induced cognitive dysfunction in experimental mice using the medicinal plant, Salvia moorcroftiana addressed most of the raised concerns. However, there are still some aspects that in my opinion need clarifications. The comments are described below:
1. In the Introduction section, please consider to switch the paragraph between lines 78-83 with the paragraph between lines 84-92. This way the introduction section will be easier to follow.

2. In Materials and Methods section, lines 139-140, rutin is abbreviated as R. However, on line 141 has been used the abbreviation RE. Please check this aspect.

3. In Materials and Methods section, line 193 ”in vivo” should be in italics. Also, at line 196 the authors mention ”…in our laboratory as well as published data elsewhere.” – please add references.

4. In Materials and Methods section, 2.9.2. subsection, line 218, please delete „…for spontaneous alternation”. Y maze test should be enough as the subsection title. Also, the fact that the animals were located in one arm is mentioned in three separate phrases (lines 224, 227 and 229). Please check and adjust this section.

5. In Materials and Methods section, 2.9.3. subsection, lines 240 – 241: ” On test day the sample phase was conducted after 60 min of the final treatment on 28th day of our study.” However, Figure 1 shows that NORT has been performed starting with day 30. Please check this inconsistency.

6. In Materials and Methods section, line 267, citation no. 36 is Bayir et al and not Elman et al.

7. In Materials and Methods section, line 276, please add the manufacturer of the ELISA kit used to assess SOD activity.

8. In Results section, Figure 2, Y axis, please delete the word ”performance” from ”Spontaneous alternation performance (%)”.

9. In Results section, 3.5.2. subsection, Table 3 please consider to represent the values for the discrimination index as a graph. This way the differences between different treatment variants will be easier to spot.

10. In Results section, 3.5.2. subsection, at line 407, it is mentioned ”the content of ACh in the hippocampus and frontal cortex of the brain.” However, in the Materials and Methods section it is not mention which brain parts has been collected. Please insert details about which brain parts were selected and used for experiments.

11. In the Results section, for Figure 2 and Figure 3, please maintain the same color shades for the experimental groups (e.g. in Figure 2 for the groups SIMo_Crd 100mg and SIMo_Crd 200mg has been used shades of grey, while in Figure 3, for the same groups, has been used shades of yellow). Also, please maintain the same order of the experimental groups in the graphs (e.g. in Figure 2 the Donepezil group is the last column in the graph, while in Figure 3 the Donepezil group is the second column in the graph). Moreover, the name of the groups are slightly different (e.g. in Figure 2 appears a ”Normal control” group, while in the Figure 3 appears a ”Control” group. Please adjust the Figures.

12. In Results section, 3.6. subsection, please express the results as graphs. Also, which brain parts has been used to produce these results (hippocampus, frontal cortex, others…)?

13. In Discussion section, lines 492-493, please delete ”spontaneous alternation” from ”Y-maze spontaneous alternation test”.

14. In Discussion section, lines 516-525, please consider to separate the discussion regarding ACh and AChE from the discussion regarding oxidative stress parameters. 

15. In Discussion section, lines 526-533, please consider to move this paragraph at the beginning of the section. 

16. In Conclusions section, line 546 ”in the brain tissues of the infected mice”, please explain.